# OpenDlign: Open-World Point Cloud Understanding with Depth-Aligned Images

**Ye Mao    Junpeng Jing**[*]    **Krystian Mikolajczyk**
Imperial College London
https://yebulabula.github.io/OpenDlign/
{ye.mao21, j.jing23, k.mikolajczyk}@imperial.ac.uk

## Abstract

Recent open-world 3D representation learning methods using Vision-Language Models (VLMs) to align 3D point clouds with image-text information have shown superior 3D zero-shot performance. However, CAD-rendered images for this alignment often lack realism and texture variation, compromising alignment robustness. Moreover, the volume discrepancy between 3D and 2D pretraining datasets highlights the need for effective strategies to transfer the representational abilities of VLMs to 3D learning. In this paper, we present OpenDlign, a novel open-world 3D model using depth-aligned images generated from a diffusion model for robust multimodal alignment. These images exhibit greater texture diversity than CAD renderings due to the stochastic nature of the diffusion model. By refining the depth map projection pipeline and designing depth-specific prompts, OpenDlign leverages rich knowledge in pre-trained VLM for 3D representation learning with streamlined fine-tuning. Our experiments show that OpenDlign achieves high zero-shot and few-shot performance on diverse 3D tasks, despite only fine-tuning 6 million parameters on a limited ShapeNet dataset. In zero-shot classification, OpenDlign surpasses previous models by 8.0% on ModelNet40 and 16.4% on OmniObject3D. Additionally, using depth-aligned images for multimodal alignment consistently enhances the performance of other state-of-the-art models.

## 1    Introduction

3D understanding, including tasks like 3D object classification [1, 2], 3D scene segmentation [3], and 3D reconstruction [4, 5], is becoming increasingly pivotal in real-world applications like augmented/virtual reality [6, 7], autonomous vehicles [8]. Traditional 3D models [1, 2, 9, 10, 11] are closed-world, recognizing only pre-defined categories and struggling with 'unseen' ones. Recent studies aim to leverage Vision-Language Models (e.g., CLIP [12]) to develop open-world 3D models that generalize beyond 'seen' 3D data, enabling zero-shot handling of various 3D tasks.

Existing open-world 3D learning methods can be classified into depth-based and point-based approaches. Depth-based methods [13, 14, 15] convert point clouds into multi-view depth maps and use the pre-trained CLIP image encoder for 3D representations. A significant challenge is the domain gap since CLIP is pre-trained on RGB images, not depth maps. To mitigate this gap, methods like [15] introduce an additional depth encoder to align depth features with the image and text features from the pre-trained CLIP encoders, as shown in Fig. 1(a). The images used for feature alignment are produced by rendering synthetic CAD models from various camera viewpoints. Point-based methods [16, 17, 18, 19, 20] directly extract 3D representations from point clouds, thereby bypassing the need for depth map projection. However, they also require an extra point encoder for feature alignment to address format disparities between images and point clouds, as shown in Fig. 1(b). Thus, employing

---

[*]Corresponding Author

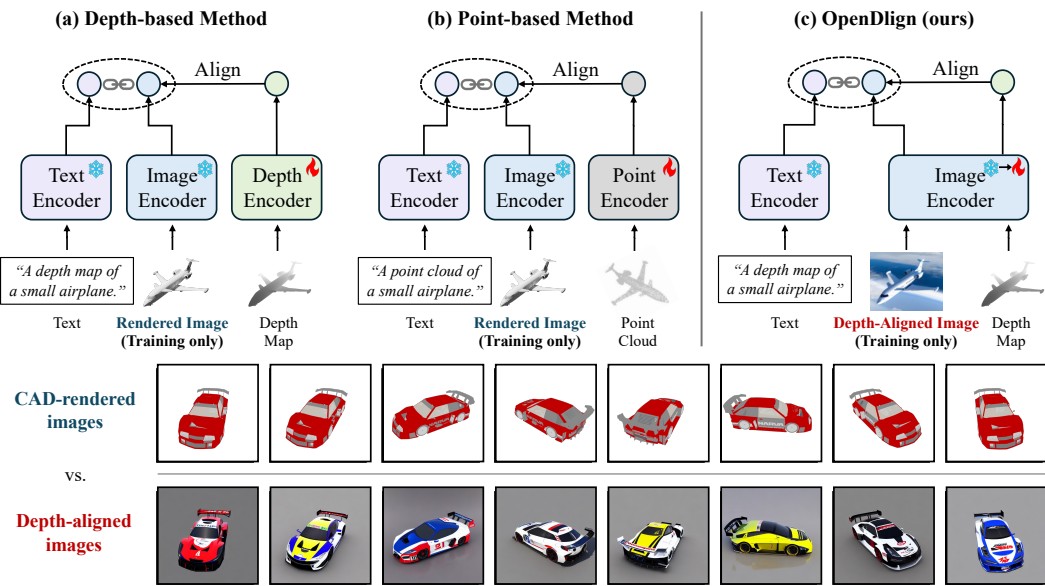

Figure 1: *Top*: Comparison of OpenDlign with traditional open-world 3D learning models. Depth-based (a) and point-based (b) methods employ additional depth or point encoders for pre-training to align with CAD-rendered images. Conversely, OpenDlign (c) fine-tunes only the image encoder, aligning with vividly colored and textured depth-aligned images for enhanced 3D representation. Both rendered and depth-aligned images are utilized solely during training. *Bottom*: Visual comparison between multi-view CAD-rendered and corresponding depth-aligned images in OpenDlign.

an extra 3D encoder for multimodal feature alignment of 3D data, CAD-rendered images, and text modalities is a standard practice in modern open-world 3D learning methods.

Despite widespread usage, we argue that CAD-rendered images fall short of providing the visual diversity and realism necessary for robust multimodal alignment. This limitation arises because CAD models in current open-source datasets [21, 22] often feature simplistic or entirely absent textures. These models also struggle to realistically simulate environmental effects like lighting, shadows, and reflections. Moreover, most CAD models maintain visual coherence across viewpoints, leading to overly consistent textures and colors from all angles. To achieve generalizable 3D representations, each image view for alignment is expected to display significant visual variations (See Fig. 1).

Additionally, pretraining an extra 3D encoder for alignment may not fully leverage the rich knowledge in CLIP for 3D understanding due to the significant volume discrepancy between 2D and 3D pre-training datasets. Mainstream 3D datasets like ShapeNet [21] and Objaverse [23] contain fewer than *1 million* synthetic 3D objects, significantly less than the vast image datasets such as DFN5B [24] and LAION-5B [25], which include around *5 billion* images used to train cutting-edge CLIP models. While direct fine-tuning of CLIP's encoders facilitates more straightforward knowledge transfer, it restricts inputs to depth maps. Yet, developing 3D representations from depth maps is currently less effective than from point clouds for two primary reasons: (1) The current widely used CLIP text prompt templates are tailored for matching with RGB images, not depth maps, and (2) the lack of a robust projection method for creating dense depth maps with smooth contours from point clouds.

In this paper, we present *OpenDlign*, the first 3D open-world framework that develops 3D representation by aligning with multi-view diffusion model-generated images, termed *depth-aligned images*. These images benefit from the stochastic nature of the diffusion model, offering greater texture diversity compared to CAD renderings while maintaining the original 3D data's geometric and semantic integrity (See Fig. 1). Remarkably, OpenDlign demonstrates competitive open-world capability by fine-tuning only 6 million parameters of the CLIP image encoder on ShapeNet [21], unleashing CLIP's vast potential in 3D learning (See Fig. 1(c)). The success of this fine-tuning stems from refining the depth map projection pipeline, developing depth-specific text prompts, and introducing a logit aggregation strategy to merge multi-view prediction results. Experimental results reveal that OpenDlign significantly outperforms previous state-of-the-art (SOTA) models on a variety

of 3D tasks, including zero-shot/few-shot classification, object detection, and cross-modal retrieval. In zero-shot classification, OpenDlign achieves accuracy improvements of 8.0% on ModelNet40 and 16.4% on OmniObject3D, the largest real-world 3D shape dataset. Additionally, depth-aligned images markedly enhance the performance of SOTA models, consistently improving results across diverse benchmarks and highlighting their transformative impact on open-world 3D learning pipelines.

The main contributions of this paper are outlined as follows:

- We introduce *depth-aligned images* as a robust alternative to CAD-rendered images for open-world 3D learning. These images, generated from point cloud-projected depth maps using a diffusion model, offer rich and realistic texture diversity across viewpoints.
- We propose a multimodal alignment framework that robustly aligns depth maps, depth-aligned images, and text through streamlined fine-tuning of the CLIP image encoder.
- We develop a contour-aware projection pipeline to produce dense and contour-preserving multi-view depth maps from point clouds.
- We present depth-specific text prompts and a logit aggregation strategy to boost OpenDlign's zero-shot capabilities and mitigate catastrophic forgetting during alignment fine-tuning.

## 2 Related Work

### 2.1 Open-World 3D Representation Learning

Vision-Language Models (VLMs) such as CLIP [12] have revolutionized 2D representation learning in open-world settings through contrastive learning with large-scale image-text pairs [26, 27, 28, 29]. Building on this, recent studies have adapted CLIP for 3D representation learning, achieving impressive performance in diverse 3D zero-shot tasks [18, 19].

PointCLIP [14], as a pioneering study, utilizes the CLIP image encoder for extracting 3D representations from depth maps of point clouds, achieving zero-shot recognition by aligning with text embeddings of semantic categories. To address CLIP's training bias towards RGB images, Zhu *et al.* [13] introduced GPT-generated 3D-specific prompts and a denser depth map projection, while CLIP2Point [15] pre-trains a depth encoder for closer alignment with CLIP's encoders. These methods derive representations from depth maps with noisy contours, causing a loss of key shape features needed for precise recognition. Moreover, their reliance on either natural image text prompts or depth-specific prompts generated by GPT-3 [30] for certain categories highlights a lack of versatility in handling diverse 3D contexts. Alternative methods [17, 18, 19, 20] avoid depth map projection by directly aligning point clouds, images, and text using specialized 3D encoders. By scaling up the dataset and encoder sizes, these methods show promise in diverse 3D tasks. However, these methods are limited by their reliance on CAD-rendered images, which have limited texture diversity across views, leading to less generalizable representations. Additionally, the smaller volume of 3D datasets compared to CLIP's training data hinders effective knowledge transfer to point cloud encoders.

In this paper, we substitute rendered images with depth-aligned images generated from a diffusion model to enhance texture diversity. We also fine-tune the CLIP image encoder for 3D representation learning instead of training a new 3D encoder from scratch, reducing the reliance on large 3D datasets.

### 2.2 Continual Learning in CLIP Fine-Tuning

Continual Learning (CL) in CLIP aims to mitigate catastrophic forgetting [31], ensuring retention of zero-shot capabilities across varied data distributions while fine-tuning to new tasks. CL methods fall into three categories: adaptive-plasticity methods [32, 33], replay methods [34, 35], and architecture-based methods [36, 37]. Adaptive-plasticity methods limit the plasticity of the essential model parameters for past tasks during fine-tuning. For instance, the IMM-Mean [38] method achieves CL by simply averaging parameters of pre-trained and fine-tuned models for inference, although its efficacy might be limited for complex tasks [39]. Replay methods leverage stored exemplars to enable CLIP to recall previously learned knowledge, while they encounter scalability challenges. Without relying on exemplars, architecture-based CL methods dynamically adjust the model's architecture to accommodate new information without losing existing knowledge [39]. In this study, we align the depth map with the RGB image by freezing the pre-trained CLIP encoder weights and

incorporating a trainable transformer-based branch for encoding depth maps, adhering to architecture-based principles. Inspired by IMM-Mean [38], we use pre-trained and fine-tuned model weights to compute classification logits from multi-view depth maps for inference.

## 3 Methodology

The OpenDlign framework, depicted in Fig. 2, starts with a contour-aware projection method that transforms point clouds into multi-view depth maps with preserved contours. These maps guide a diffusion model to produce depth-aligned images with varied colors and textures, maintaining consistent geometry with the depth maps. OpenDlign then aligns features from depth maps and depth-aligned images by fine-tuning a transformer block connected to the pre-trained image encoder. The goal is to align feature embeddings from depth maps and corresponding depth-aligned images using contrastive learning. At inference, 3D representations are composed of embeddings from multi-view depth maps, derived using both fine-tuned and pre-trained encoder states. These embeddings are matched with depth-specific text embeddings, which capture the semantic and visual properties of the depth maps, to generate multi-view logits. These logits are then aggregated to facilitate label prediction in a zero-shot setting. In few-shot scenarios, these embeddings can be further refined with a logistic regressor. Detailed training and model implementation are provided in Appendix A.2.

### 3.1 Contour-Aware Depth Map Projection

The contour-aware projection method transforms the input point cloud into multi-view depth maps with clear contours. Inspired by the pipeline in [13], this method involves four main steps: Quantize, Densify, Smooth, and Squeeze.

In the **Quantize** step, for the $i^{\text{th}}$ view of point cloud $P_i$, the 3D coordinates $(x, y, z) \in P_i$ are normalized to $[0, 1]$ and mapped onto a discrete grid $G \in \mathbb{R}^{H \times W \times B}$, where $H$ and $W$ correspond to the dimensions required by the CLIP image encoder, and $B$ is a pre-defined depth dimension. Next, the **Densify** step enhances $G$ by updating each voxel to the maximum value within its $7 \times 7 \times 7$ neighborhood, yielding a denser map $G'$. Subsequently, the **Smooth** step applies bilateral filtering to each voxel $v_i$ in $G'$, adjusting its intensity $I_{v_i}$ to $I'_{v_i}$ using:

$$I'_{v_i} = \frac{1}{W_v} \sum_{v_j \in S} G_{\sigma_1}(\|v_i - v_j\|) G_{\sigma_2}(|I_{v_i} - I_{v_j}|) I_{v_j} \tag{1}$$

where $W_v = \sum_{v_j \in S} G_{\sigma_1}(\|v_i - v_j\|) G_{\sigma_2}(|I_{v_i} - I_{v_j}|)$ is the normalization factor that ensures voxel weights sum to 1.0. The Gaussian functions $G_{\sigma_1}$ and $G_{\sigma_2}$ adjust the influence of each neighboring voxel $v_j$ within the $5 \times 5 \times 5$ kernel from set $S$ around $v_i$, based on spatial and intensity differences, enhancing contour sharpness and reducing jagged edges in $G'$. Finally, the **Squeeze** step applies the minimal pooling on the depth channel of the smoothed $G'$, then triples the output to mimic RGB intensity, producing the final depth map $D \in \mathbb{R}^{H \times W \times 3}$.

### 3.2 Depth-Aligned Image Generation

A total of **524,700** depth-aligned images are generated from ShapeNet [21], a leading public 3D CAD dataset containing approximately 52,470 models, each annotated with semantic metadata. To generate these images, a point cloud of 10,000 points is sampled from each CAD model, aligning with prior experimental protocols [18, 17]. For each point cloud, 10 views of depth maps are projected using the proposed contour-aware projection method. Subsequently, the ControlNet v1.1 [40] depth model produces depth-aligned images for each contour-aware depth map view, using the CAD model's metadata as text input and the inverse of the depth map $\frac{1}{D}$ as image generation control. This approach ensures that the generated images remain consistent with the depth maps both geometrically and semantically, while also adding texture diversity across different views. ControlNet utilizes $\frac{1}{D}$ instead of $D$ for controlling image outputs because it is predominantly pre-trained on depth images where brighter regions indicate closer proximity. Please refer to Appendix A.2 for the positive and negative prompts used in ControlNet to achieve high-fidelity depth-aligned image generation.

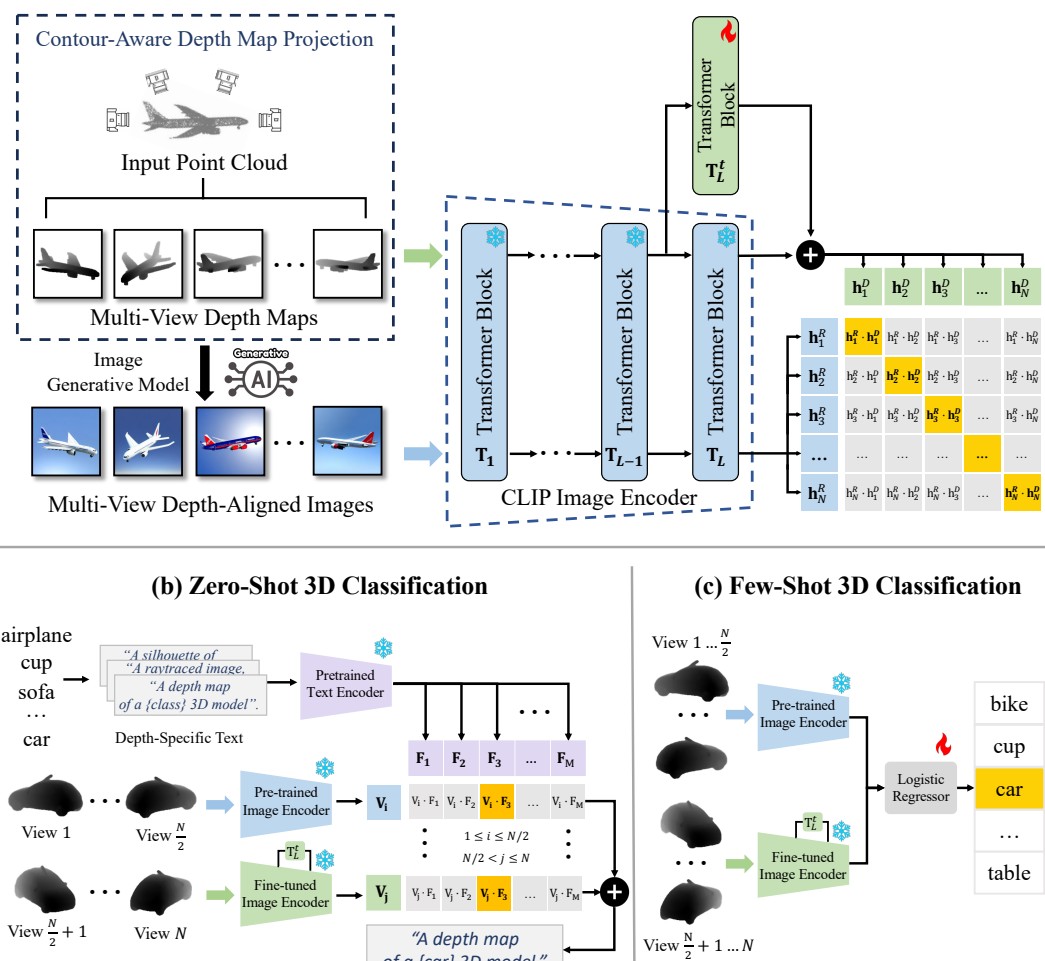

Figure 2: Overview of OpenDlign. In (a), OpenDlign converts point clouds into multi-view depth maps using a contour-aware projection, which then helps generate depth-aligned RGB images with diverse textures, geometrically and semantically aligned with the maps. A transformer block, residually connected to the CLIP image encoder, is fine-tuned to align depth maps with depth-aligned images for robust 3D representation. For zero-shot classification (b), OpenDlign aggregates multi-view logits from both pre-trained and fine-tuned encoders for label prediction. For few-shot classification (c), it employs a logistic regressor trained on multi-view features from the encoders.

## 3.3 Multimodal Representation Alignment

OpenDlign aligns depth maps and depth-aligned images by fine-tuning a transformer block residually linked to the last block of the CLIP image encoder, using contrastive learning. With CLIP pre-trained to align images and text, OpenDlign implicitly aligns depth maps with the shared image-text space.

**Multimodal Feature Extraction.** Given a 3D point cloud input, let $D = \{D_i\}_{i=1}^N$ represent the set of its $N$ projected depth map views, and $R = \{R_i\}_{i=1}^N$ the corresponding set of depth-aligned images. Each image $R_i$ is encoded through $L$ layers of a pre-trained CLIP image encoder, $\{T_l(\cdot)\}_{l=1}^L$, to obtain feature representations $I_i^R = T_{1...L}(R_i)$. Each depth map $D_i$ is processed up to layer $T_{L-1}$, yielding preliminary features $T_{1...L-1}(D_i)$. Subsequently, these depth features are passed through the frozen layer $T_L$ and its trainable counterpart $T_L^t$, where only the attention layers for spatial interaction in $T_L^t$ are trainable, as inspired by [41]. This process produces the feature for the $i_{th}$ depth map view

$I_i^D = \mathrm{T}_{1\ldots L}(D_i) + \mathrm{T}_L^t(\mathrm{T}_{1\ldots L-1}(D_i))$. The final feature vectors for multi-view depth maps $D$ and depth-aligned images $R$ are $\mathbf{h}^D = \frac{1}{N}\sum_{i=1}^{N}\|I_i^D\|$ and $\mathbf{h}^R = \frac{1}{N}\sum_{i=1}^{N}\|I_i^R\|$, respectively.

**Loss Functions.** The alignment of $\mathbf{h}^D$ and $\mathbf{h}^R$ is achieved by minimizing a composite loss function, comprising the contrastive loss $\mathcal{L}_{\text{cont}}$ and the feature distance loss $\mathcal{L}_{\text{dist}}$, defined as:

$$\mathcal{L}_{\text{total}} = \underbrace{\sum_{(i,j)} -\frac{1}{2}\log\frac{\exp\left(\mathbf{h}_i^D\mathbf{h}_j^R/\tau\right)}{\sum_k \exp\left(\mathbf{h}_i^D\mathbf{h}_k^R/\tau\right)} - \frac{1}{2}\log\frac{\exp\left(\mathbf{h}_i^D\mathbf{h}_j^R/\tau\right)}{\sum_k \exp\left(\mathbf{h}_k^D\mathbf{h}_j^R/\tau\right)}}_{\mathcal{L}_{\text{cont}}} + \underbrace{\sum_{(i,j)}\|\mathbf{h}_i^D - \mathbf{h}_j^R\|_2}_{\mathcal{L}_{\text{dist}}} \quad (2)$$

In each training batch, $(\mathbf{h}_i^D, \mathbf{h}_i^R)$ is a positive pair, while $(\mathbf{h}_i^D, \mathbf{h}_k^R)$ and $(\mathbf{h}_k^D, \mathbf{h}_j^R)$ are negative pairs where $k \neq i, j$. $\tau$ is a learnable temperature parameter. The contrastive loss enables learning robust representations by maximizing similarity in positive pairs and minimizing it in negative pairs [42, 43].

### 3.4 3D Zero-Shot Transfer

The alignment between depth maps and depth-aligned images facilitates 3D zero-shot classification by aggregating multi-view classification logits. Each logit represents the similarity between single-view depth features and text features of category candidates, as illustrated in Fig. 2(b).

**Depth-Specific Text Generation.** Depth-specific prompt templates are developed by augmenting a base set of 80 text prompts, initially designed for ImageNet classification[2], with depth-related keywords such as "depth map", "raytraced image", and "silhouette of [CLASS]". These keywords direct OpenDlign's attention to geometric details and contours rather than color or texture. The CLIP Interrogator [44] is a prompt engineering tool that combines CLIP and BLIP [45] to select optimal text prompts for specific images. To identify these keywords, this tool identifies the top 10 prompts that match depth maps from the ShapeNet dataset [21], chosen as targeted keywords. For zero-shot inference, we employ our depth-specific templates to generate 80 text descriptions for each label $l$. These descriptions $\{t_i\}_{i=1}^{80}$ are encoded by a texture encoder $F(\cdot)$, normalized, and then merged into a unified text feature $F_l$ via average pooling, calculated as $\frac{1}{80}\sum_{i=1}^{80}\|F(t_i)\|$.

**Multi-View Logits Aggregation.** To calculate classification logits, we first gather visual features from multi-view depth maps $\{V_i\}_{i=1}^{N}$, aiming to align with depth-specific text features of $M$ candidate labels $\mathbf{F} = \{F_i\}_{i=1}^{M}$. The feature extraction utilizes a dual-encoder strategy: the first half of the views $\{V_i\}_{i=1}^{N/2}$ utilize a pre-trained CLIP image encoder, while the second half of the views $\{V_i\}_{i=N/2+1}^{N}$ employs a fine-tuned encoder. The strategy ensures that OpenDlign maintains its capability to recognize previously identifiable depth maps after learning multimodal alignment via fine-tuning. As shown in Fig. 2(b), the logit for a single depth map view is the product of $V_i$ and $\mathbf{F}$, with the overall classification logit being the sum of logits across all views, calculated as $\sum_{i=1}^{N} V_i\mathbf{F}^T$.

## 4 Experiments

### 4.1 Zero-Shot 3D Classification

We first evaluated OpenDlign under the zero-shot shape classification task on four benchmark datasets: ModelNet40 [46], ScanObjectNN [47], OmniObject3D [48], and Objaverse-LVIS [23]. ModelNet40 offers synthetic 3D CAD models in 40 categories. ScanObjectNN provides real-scanned objects in 15 categories from OBJ_ONLY version. OmniObject3D, the largest, includes 5,911 real-scanned objects in 216 categories, well-suited for fine-grained, real-world classification evaluation. Objaverse-LVIS contains 1,156 categories for evaluating long-tail classification. Point cloud sizes are 10,000 points for ModelNet40 and Objaverse-LVIS, 2,048 for ScanObjectNN, and 4,096 for OmniObject3D. OpenDlign was compared against existing methods, including three depth-based methods: PointCLIP [14], PointCLIP V2 [13], and CLIP2Point [15], and three point-based methods: ULIP [17], OpenShape [18], and TAMM [20]. To investigate if depth-aligned images consistently enhance the representational abilities of other 3D open-world methods, we retrained all OpenShape and TAMM variants using their original CAD-rendered images and some depth-aligned images. Note

---

[2]Text Prompts for ImageNet: ImageNet Prompt Engineering.

Table 1: Zero-shot classification results on ModelNet40 [46], ScanObjectNN [47] and OmniObject3D [48]. The best-performing results are presented in bold, while the second-best results are underlined. Our models are highlighted in blue.

| Training Source | 3D Open-World Methods | CLIP Variant | ModelNet40 [46] | | | ScanObjectNN [47] | | | OmniObject3D [48] | | |
|---|---|---|---|---|---|---|---|---|---|---|---|
| | | | Top1 | Top3 | Top5 | Top1 | Top3 | Top5 | Top1 | Top3 | Top5 |
| 2D inferences No Training | PointCLIP [14] | ResNet-50 | 19.3 | 28.6 | 34.8 | 10.5 | 20.8 | 30.6 | 0.3 | 1.0 | 1.8 |
| | PointCLIP V2 [13] | ViT-B-16 | 63.6 | 77.9 | 85.0 | 42.2 | 63.3 | 74.5 | 3.9 | 9.6 | 14.4 |
| ShapeNet | CLIP2Point [15] | ViT-B-32 | 49.5 | 71.3 | 81.2 | 25.5 | 44.6 | 59.4 | 1.4 | 3.7 | 7.1 |
| | ULIP-PointBERT [17] | SLIP [50] | 60.4 | 79.0 | 84.4 | 51.5 | 71.1 | 80.2 | 8.4 | 15.2 | 19.7 |
| | OpenShape-SparseConv [18] | ViT-bigG-14 | 72.9 | 87.2 | 93.0 | 52.7 | 72.7 | 83.6 | 13.7 | 24.2 | 30.0 |
| | OpenShape-PointBERT [18] | ViT-bigG-14 | 70.3 | 86.9 | 91.3 | 51.3 | 69.4 | 78.4 | 13.0 | 23.3 | 29.4 |
| | TAMM-SparseConv [20] | ViT-bigG-14 | 74.6 | 88.2 | 94.0 | 57.9 | 75.3 | 83.1 | - | - | - |
| | TAMM-PointBERT [20] | ViT-bigG-14 | 73.1 | 88.5 | 91.9 | 54.8 | 74.5 | 83.3 | 14.9 | 26.2 | 33.4 |
| | OpenShape-SparseConv (+dlign) | ViT-bigG-14 | 74.9 | 89.5 | 94.1 | 56.3 | 75.2 | **85.4** | 15.0 | 26.1 | 32.8 |
| | OpenShape-PointBERT (+dlign) | ViT-bigG-14 | 73.7 | 87.1 | 91.3 | 52.7 | 72.4 | 82.6 | 13.4 | 23.7 | 29.9 |
| | TAMM-PointBERT (+dlign) | ViT-bigG-14 | 73.7 | 89.1 | 92.2 | 57.3 | 73.6 | 82.3 | 15.8 | 27.4 | 33.0 |
| | OpenDlign-B32 | ViT-B-32 | 68.4 | 86.4 | 92.6 | 46.7 | 72.0 | 83.0 | 17.3 | 29.2 | 36.3 |
| | OpenDlign-B16 | ViT-B-16 | 74.2 | 90.5 | 95.4 | 49.3 | 74.0 | 84.4 | 23.2 | 37.5 | 44.3 |
| | OpenDlign-L | ViT-L-14 | 77.8 | 93.1 | 96.4 | 52.1 | 74.6 | 82.8 | 27.5 | 41.3 | 47.8 |
| | **OpenDlign** | **ViT-H-14** | **82.6** | **96.2** | **98.4** | **59.5** | **76.8** | 83.7 | **31.3** | **46.7** | **53.2** |
| Ensemble | OpenShape-SparseConv [18] | ViT-bigG-14 | 83.4 | 95.6 | 97.8 | 56.7 | 78.9 | 88.6 | 33.7 | 49.3 | 57.4 |
| | OpenShape-PointBERT [18] | ViT-bigG-14 | 84.4 | 96.5 | 98.0 | 52.2 | 79.7 | 88.7 | 34.0 | 49.7 | 57.9 |
| | TAMM-PointBERT [20] | ViT-bigG-14 | 85.0 | 96.6 | 98.1 | 55.7 | 80.7 | 88.9 | 37.1 | 53.5 | 61.8 |
| | TAMM-SparseConv [20] | ViT-bigG-14 | 85.4 | 96.4 | 98.1 | 58.5 | 81.3 | 89.5 | - | - | - |
| | OpenShape-SparseConv (+dlign) | ViT-bigG-14 | 85.0 | 96.1 | 97.9 | 56.2 | 78.5 | 87.8 | 34.1 | 50.5 | 58.5 |
| | OpenShape-PointBERT (+dlign) | ViT-bigG-14 | 85.4 | 96.6 | 98.2 | 51.1 | 77.4 | 88.2 | 35.6 | 50.4 | 57.9 |
| | **TAMM-PointBERT (+dlign)** | **ViT-bigG-14** | **86.2** | **96.6** | 97.5 | **60.5** | **82.5** | **90.4** | **37.5** | **54.9** | **62.1** |

that the depth-aligned images for all model variants were exclusively generated from ShapeNet [21], while the pretraining CAD-rendered images could come from ShapeNet or a large-scale ensemble dataset [18] that includes Objaverse [23], ShapeNet [21], 3D-Future [49], and ABO [22]. Furthermore, we evaluated OpenDlign's scalability by training it with various CLIP variants.

Table 6 shows OpenDlign substantially outperforms existing methods trained on ShapeNet across all benchmarks, exceeding the previous best, TAMM-SparseConv, by margins of 8.0% on ModelNet40, 1.6% on ScanObjectNN, and 16.4% on OmniObject3D in Top1 accuracy. Appendix A.5 Table 7 demonstrates OpenDlign's excellence in handling long-tail categories by outperforming previous methods by 21.3% on the Objaverse-LVIS dataset. OpenDlign also outperforms the top depth-based method, PointCLIP V2, by 19% on ModelNet40 and 27.4% on OmniObject3D. Notably, OpenDlign outshines all methods pre-trained on the ensemble dataset in the ScanObject3D benchmark. Additionally, its performance increases linearly with the complexity of CLIP variants, outperforming most baselines on ModelNet40 and OmniObject3D, even using the lighter ViT-B-16 CLIP model. Remarkably, using depth-aligned images (+dlign) consistently boosts the performance of OpenShape and TAMM variants pre-trained on the ShapeNet dataset across all benchmarks. Despite depth-aligned images being available only for ShapeNet, which comprises just 10% of the ensemble dataset, TAMM-PointBERT (+dlign) shows a 4.8% increase in Top1 accuracy on the ScanObjectNN dataset, and OpenShape-PointBERT (+dlign) records a 1.6% increase on the OmniObject3D. Note that for parts of the ensemble dataset without depth-aligned images, we used CAD-rendered images instead.

Table 2: Few-shot classification results on ModelNet40 [46], ScanObjectNN [47] and OmniObject3D [48]. Our results are averaged over 10 random seeds.

| Model | ModelNet40 [46] | | | | | ScanObjectNN [47] | | | | | OmniObject3D [48] | | | | |
|---|---|---|---|---|---|---|---|---|---|---|---|---|---|---|---|
| | 1-Shot | 2-Shot | 4-Shot | 8-Shot | 16-Shot | 1-Shot | 2-Shot | 4-Shot | 8-Shot | 16-Shot | 1-Shot | 2-Shot | 4-Shot | 8-Shot | 16-Shot |
| ULIP-PointBERT [17] | 54.4 | 64.3 | 74.1 | 79.3 | 81.3 | 46.7 | 55.1 | 62.5 | 70.7 | 73.9 | 37.5 | 41.2 | 44.1 | 49.7 | 53.4 |
| OpenShape-PointBERT [18] | 57.5 | 70.1 | 76.5 | 80.4 | 82.1 | 47.9 | 55.6 | 62.7 | 67.0 | 72.0 | 34.5 | 34.1 | 37.8 | 41.9 | 45.6 |
| OpenShape-SparseConv [18] | 62.8 | 72.0 | 78.9 | 82.9 | 85.7 | 47.3 | 56.3 | 64.5 | 68.2 | 74.0 | 36.0 | 37.0 | 41.5 | 44.7 | 48.6 |
| TAMM-PointBERT [20] | 62.4 | 73.3 | 81.7 | 83.8 | 85.9 | 48.2 | 57.1 | 63.6 | 72.1 | 76.5 | 38.9 | 41.6 | 46.3 | 50.1 | 54.2 |
| **OpenDlign (ours)** | **65.6** | **73.9** | **82.9** | **85.5** | **87.6** | **48.9** | **58.5** | **67.9** | **74.2** | **79.0** | **42.1** | **46.9** | **55.1** | **61.9** | **65.8** |

## 4.2 Few-Shot 3D Classification

We then assessed OpenDlign's few-shot classification capability by training a logistic regressor with linear probing on features from $N$-shot, 10-view depth maps. Similar to the zero-shot scenario, we extracted multi-view features using both fine-tuned and pre-trained OpenDlign encoders. At

Table 3: Zero-shot 3D object detection results on ScanNet V2 [51].

| | Method | Mean | Cabinet | Bed | Chair | Sofa | Table | Door | Window | Counter | Desk | Sink | Bathtub |
|---|---|---|---|---|---|---|---|---|---|---|---|---|---|
| $AP_{25}$ | PointCLIP [14] | 6.00 | 3.99 | 4.82 | 45.16 | 4.82 | 7.36 | 4.62 | 2.19 | 1.02 | 4.00 | 13.40 | 6.46 |
| | PointCLIP V2 [13] | 18.97 | 19.32 | 20.98 | 61.89 | 15.55 | 23.78 | 13.22 | 17.42 | 12.43 | 21.43 | 14.54 | 16.77 |
| | **OpenDlign (ours)** | **50.72** | **38.91** | **67.27** | **86.33** | **72.01** | **58.72** | **44.58** | **32.07** | **50.49** | **62.04** | **51.98** | **64.29** |
| $AP_{50}$ | PointCLIP [14] | 4.76 | 1.67 | 4.33 | 39.53 | 3.65 | 5.97 | 2.61 | 0.52 | 0.42 | 2.45 | 5.27 | 1.31 |
| | PointCLIP V2 [13] | 11.53 | 10.43 | 13.54 | 41.23 | 6.60 | 15.21 | 6.23 | 11.35 | 6.23 | 10.84 | 11.43 | 10.14 |
| | **OpenDlign (ours)** | **37.97** | **17.04** | **66.68** | **73.92** | **54.96** | **50.03** | **24.73** | **12.84** | **20.44** | **41.64** | **34.17** | **64.29** |

inference, the regressor aggregates logits from 10 views to predict the final label. We compared OpenDlign with ULIP [17], OpenShape [18], and TAMM [20], which extract features for training regressor from their point encoders pre-trained on ShapeNet. Table 2 shows OpenDlign outperforms all baselines across varied few-shot scenarios with 1 to 16 training samples per class. OpenDlign significantly outperforms the leading baseline on the OmniObject3D dataset, exceeding it by 8.8% and 11.8% in the 4-shot and 8-shot classification, respectively. See Appendix A.5 for more results.

## 4.3 Zero-Shot 3D Object Detection

OpenDlign's capability in zero-shot 3D object detection was evaluated on the ScanNet V2 dataset [51], which contains richly annotated 3D indoor scenes in 18 object categories. Following the PointCLIP V2 methodology [13], we started with the pre-trained 3DETR-m [11] model to identify 3D regions of interest, delineate 3D bounding boxes, and extract points within each box. Finally, we applied OpenDlign to these points to generate our predictions. Table 3 illustrates OpenDlign's zero-shot detection prowess using mean Average Precision (mAP) at IoU thresholds of 0.25 and 0.5, achieving scores of 50.72% and 37.97%, respectively. It significantly outperforms PointCLIP V2 by more than 31.75% and 26.44%. Remarkably, OpenDlign can detect the 'Sofa' shape with an $AP_{50}$ of 54.96%, whereas PointCLIP and V2 score below 10%, demonstrating OpenDlign's superior ability to extract robust 3D representations from sparse and noisy point clouds in real-world indoor scenes.

## 4.4 Cross-Modal Retrieval

3D shapes were retrieved by computing the cosine similarity between the embeddings of a query and those generated by OpenDlign, followed by a k-nearest neighbors (kNN) analysis to find the most similar shapes. Fig. 3 showcases OpenDlign's ability to match 3D shapes to image and text queries. Column (a) shows its precision in distinguishing sub-categories like grand versus upright pianos from image queries. Column (b) demonstrates successful shape retrieval using distinctive text descriptions like "Batmobile armored". Notably, averaging image and text query embeddings allows OpenDlign to find shapes that combine elements of both queries. For instance, merging a running horse image with the text "man" retrieves both a centaur and a running man, as shown in Fig. 3(c). Similarly, combining a house image with "tree" retrieves a treehouse. See Appendix A.6 for more results.

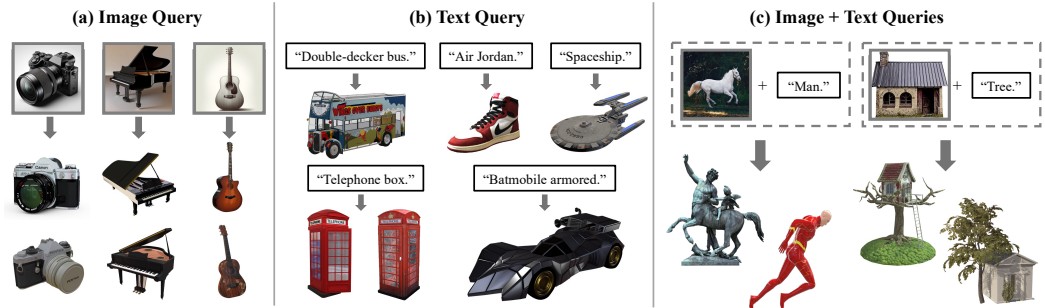

Figure 3: 3D shape retrieval results. (a) Two most similar shapes for each image query. (b) Most similar shapes for each text query. (c) Two most similar shapes for combined image and text queries.

Table 4: Ablation study for OpenDlign on ModelNet40 [46] and ScanObjectNN [47]. Accuracy improvements over the baseline (first-row) are highlighted in green.

| Contour-Aware Projection | Multimodal Alignment | Depth-Specific Texts | Logits Aggregation | ModelNet40 [46] | | | ScanObjectNN [47] | | |
|---|---|---|---|---|---|---|---|---|---|
| | | | | Top1 | Top3 | Top5 | Top1 | Top3 | Top5 |
| ✗ | ✗ | ✗ | ✗ | 59.7 | 79.6 | 86.3 | 42.8 | 66.7 | 78.4 |
| ✓ | ✗ | ✗ | ✗ | 68.8 (+9.1) | 85.8 (+6.2) | 91.6 (+5.3) | 44.6 (+1.8) | 68.3 (+1.6) | 78.9 (+0.5) |
| ✓ | ✓ | ✗ | ✗ | 79.2 (+19.5) | 94.4 (+14.8) | 97.6 (+11.3) | 56.9 (+14.1) | 75.5 (+8.8) | 83.8 (+5.4) |
| ✓ | ✗ | ✓ | ✗ | 75.9 (+16.2) | 91.0 (+11.4) | 95.4 (+9.1) | 49.3 (+6.5) | 69.8 (+3.1) | 79.2 (+0.8) |
| ✓ | ✓ | ✓ | ✗ | 80.2 (+20.5) | 95.3 (+15.7) | 97.7 (+11.4) | 58.1 (+15.3) | 75.2 (+8.5) | 84.2 (+5.8) |
| ✓ | ✓ | ✗ | ✓ | 81.0 (+21.3) | 95.2 (+15.6) | 97.6 (+11.3) | 56.8 (+14.0) | 74.6 (+7.9) | 81.6 (+3.2) |
| ✓ | ✓ | ✓ | ✓ | 82.6 (+22.9) | 96.2 (+16.6) | 98.4 (+12.1) | 59.5 (+16.7) | 76.8 (+10.1) | 83.7 (+5.3) |

## 4.5 Ablation Study

Ablation studies were conducted on zero-shot classification benchmarks to assess the contribution of each component in OpenDlign. Consistently, all OpenDlign variants used in these studies employed OpenCLIP-ViT-H-14 as their backbone. ShapeNet was the default training dataset for all models.

**Contour-Aware Projection.** Replacing PointCLIP V2's projection pipeline [13] with our contour-aware version enables a pre-trained CLIP to achieve 68.8% zero-shot accuracy on ModelNet40, outperforming several baselines that require extra training (See Table 4). This indicates that CLIP can understand RGB images and depth maps when shape features are highlighted.

**Effect of Alignment with Depth-Aligned Images.** Table 4 shows that aligning depth maps with depth-aligned images (i.e., depth-dlign) significantly boosts performance, improving Top1 accuracy by around 10% on ScanObjectNN, with or without depth-specific prompts. This indicates that depth-d alignment effectively transfers CLIP's rich knowledge to interpret depth maps.

Further analysis compared depth-dlign alignment against three alternatives: depth-rendCAD (aligning depth maps with CAD-rendered RGB images), dlign-text & depth (aligning depth-aligned images with text before depth-dlign alignment), and depth-text & dlign (simultaneous alignment of depth maps with text and depth-aligned images). Table 5 shows depth-dlign outperforming depth-rendCAD by 6.8% on the ScanObjectNN dataset, confirming concerns that alignment with rendered images may lead to overfitting on specific 3D shapes. Moreover, dlign-text & depth performs worst, suggesting that pre-aligning depth-aligned images with text compromises CLIP's ability to generate robust image representations, thus affecting subsequent depth-dlign alignment efficacy. The superior performance of depth-dlign on ModelNet40 and OmniObject3D compared to depth-text & dlign shows that aligning depth maps with depth-aligned images indirectly aligns with text, making additional text alignment unnecessary and potentially limiting OpenDlign's generalization.

**Depth-Specific Texts.** Table 4 shows that depth-specific prompts enhance OpenDlign's performance, regardless of using multimodal alignment or logit aggregation. This indicates that some recognition inaccuracies arise from processing input data as typical RGB images instead of depth maps.

**Logits Aggregation.** Results in Table 4 show that multi-view logit aggregation improves zero-shot classification on all datasets by combining logits from pre-trained and fine-tuned encoders. This approach effectively mitigates the catastrophic forgetting problem in OpenDlign's multimodal alignment, enabling it to recognize 3D objects identifiable by both pre-trained CLIP and OpenDlign.

Table 5: Ablation study on various alignment strategies. Aligning with text modality was achieved by fine-tuning the image encoder.

| Alignment Strategy | MNet40 [46] | | ScanNN [47] | | Omni3D [48] | |
|---|---|---|---|---|---|---|
| | Top1 | Top5 | Top1 | Top5 | Top1 | Top5 |
| depth-rendCAD | 78.8 | 96.8 | 52.7 | 82.5 | 29.4 | 51.8 |
| dlign-text & depth | 78.6 | 96.4 | 51.1 | 79.6 | 29.1 | 51.6 |
| depth-text & dlign | 79.4 | 98.0 | 60.7 | 86.0 | 29.5 | 52.7 |
| **depth-dlign (ours)** | 82.6 | 98.4 | 59.5 | 83.7 | 31.3 | 53.2 |

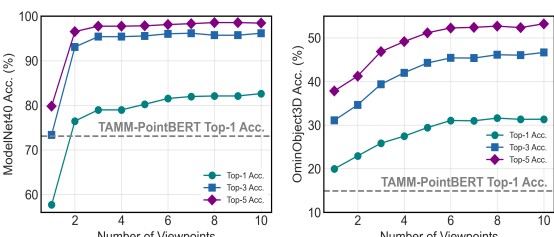

Figure 4: Effect of the number of views on OpenDlign's zero-shot performance.

**Varying Number of Views.** OpenDlign, like other depth-based methods, requires extracting multiple embeddings from multi-view depth maps for zero-shot inference. Fig. 7 shows that OpenDlign's zero-shot accuracy on ModelNet40 and OmniObject3D improves with more depth map views. Notably, OpenDlign achieves top performance, comparable to TAMM-PointBERT, with just two views, balancing latency and effective zero-shot classification.

## 5 Conclusion

In this study, we introduce OpenDlign, an open-world framework that learns robust 3D representations from multi-view depth maps by efficiently fine-tuning with depth-aligned images, which are more visually diverse than CAD-rendered images. The effectiveness of OpenDlign is validated on various 3D zero-shot and few-shot tasks. We also show that depth-aligned images consistently enhance the performance of existing 3D open-world methods. Future work will explore the application of depth-aligned images in designing open-world models for 3D scenes (See Appendix A.1).

**Limitations.** Due to limited computational resources, we cannot generate depth-aligned images from the largest 3D dataset [23], containing around 1 million 3D objects. Retraining 3D open-world models with billions of parameters using these images is also too expensive. Moreover, a data filtering strategy is needed to remove low-quality depth-aligned images (See details in Appendix A.6).

## Acknowledgement

We extend our gratitude to Dylan Auty, Maojun Zhang, Ranran Huang, Jiangnan Ye, Ziyang Chen, and Chengzu Li for their valuable discussions and insightful feedback on the early drafts of this work. This research was supported by the Imperial College President's PhD Scholarships 2023.

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

## A  Appendix / supplemental material

### A.1  Broader Impact

The broader impact of OpenDlign, a novel 3D open-world framework, includes both potential benefits and negative impacts associated with its deployment and use. Some considerations are unique to OpenDlign due to its requirement to generate multi-view depth-aligned images, while others are similar to those of existing 3D open-world methods like OpenShape [18], and TAMM [20].

**Positive Impacts.** While OpenDlign's primary focus is on 3D shape understanding, it shows significant potential for 3D scene understanding. Existing 3D open-world methods are limited by relying on CAD-rendered images for alignment, as CAD models are typically available only for individual objects, not entire scenes. In contrast, OpenDlign generates depth-aligned images using depth maps instead of CAD models. These depth maps can be produced from point clouds obtained either by sampling from CAD models or using real-scanned point clouds collected via LiDAR. Using real-scanned point clouds allows OpenDlign to have richly textured scene images for multimodal alignment, facilitating the learning of robust 3D scene representations.

**Negative Impacts.** Biases present in CLIP can be transferred to OpenDlign, potentially resulting in biased outcomes or unfair 3D representations of diverse content. Additionally, the generated

depth-aligned images may exhibit biased and stereotypical traits due to inherent biases in the training data of ControlNet. OpenDlign requires generating a large number of multi-view depth-aligned images for multimodal alignment using a diffusion model, raising concerns about energy consumption. Furthermore, if a larger CLIP backbone is released in the future, the increasing fine-tuning costs of OpenDlign could exacerbate these energy concerns.

In summary, OpenDlign shares common concerns with existing 3D open-world frameworks. However, its positive impacts are unique and hold significant potential to transform the design of future 3D open-world understanding pipelines.

## A.2   Implementation Details

**Training Details.** OpenDlign was implemented in PyTorch, utilizing the image-text encoders from OpenCLIP-ViT-H-14, pre-trained on the DFN5B dataset [24]. OpenDlign uses contour-aware projections to transform point clouds into depth maps with dimensions of $224 \times 224 \times 3$, creating 10 views along the x-axis, ranging from 30 to 330 degrees at intervals of 30 degrees between each pair of views. The multimodal alignment was achieved by fine-tuning 10 epochs on an A100-80 GB GPU, employing the AdamW optimizer and the OneCycle scheduler with a peak learning rate of $3 \times 10^{-4}$ and a batch size of 128. Since we precache the text and image CLIP embeddings of all shapes, training is significantly accelerated, achieving convergence in about 10 hours.

**Depth-Aligned Image Generation Details.** We employed the ControlNet v1.1 model [40], pre-trained on depth maps, with settings of 20 DDIM steps, a 9.0 guidance scale, and a 1.0 control strength for generating depth-aligned images. The prompts for generating high-quality, noise-free depth-aligned images are detailed in Table 6. We discovered that ControlNet [40] is more sensitive to jagged edges in the conditional depth map than Vision-Language Models. Our contour-aware projection reduces these jagged edges using bilateral filtering in each depth channel. However, the edges may reappear during the channel squeezing step. To combat this, we applied a $7 \times 7$ kernel median filter to each projected depth map to further diminish the edges. The output images, initially sized at $256 \times 256 \times 3$, were then downsampled to match the depth maps' dimensions. The entire generation process for ShapeNet [21] spanned 16 days on 8 RTX 6000 GPUs.

Table 6: Main, positive, and negative prompts guide depth-aligned image generation. Metadata textually describes the semantic information of the 3D data.

| Main Prompt | "A realistic {metadata}." |
|---|---|
| Positive Prompts | "best quality, extremely realistic, very professional, extremely detailed, sharp edge, normal, complete." |
| Negative Prompts | "low-resolution, very blurry, unrealistic, worst quality, deep depth of field, large depth of field, distorted, cropped, unusual, warped, incomplete." |

## A.3   Discussion

**How to ensure multi-view geometry consistency in depth-aligned images?** There is no geometric inconsistency because depth-aligned images are generated view by view, each paired with its respective depth map. ControlNet's [40] conditional control ensures the generated images maintain the same shape and pose as the input depth map, while a text prompt preserves object identity. Since these depth maps originate from the same 3D point cloud, the resulting images remain geometrically consistent across views. Fig. 5 visually demonstrates the generated images are geometry-consistent.

**Why does increasing texture diversity in multi-view images positively impact open-world 3D learning?** First, more effective knowledge transfer is achieved by leveraging inconsistent textures in multi-view generated images, allowing depth maps to align with diverse sets of RGB images, which enhances the transfer of rich 2D knowledge embedded in CLIP to 3D representation learning. Second, generating geometry-consistent but texture-inconsistent multi-view images benefits 3D representation learning by focusing on invariant features like object shape, size, and contour, which remain robust to texture variations. Third, similar to previous methods [15, 17, 18], OpenDlign pairs a single-view depth map with a single-view RGB image for contrastive learning, treating each view independently and eliminating the need for texture consistency across views. Lastly, texture features

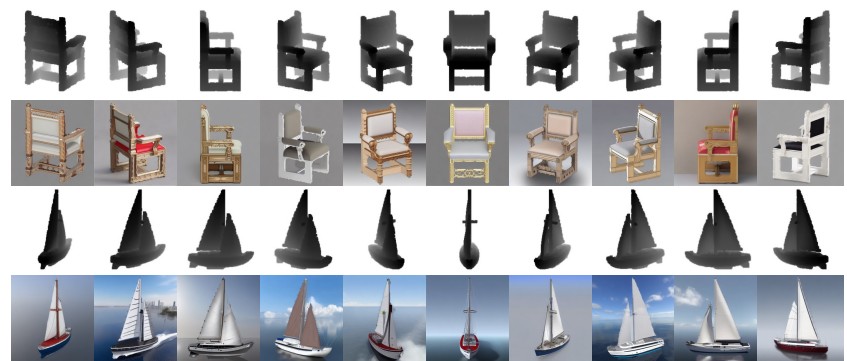

Figure 5: Examples of multi-view depth maps and their corresponding depth-aligned images.

are generally unreliable in both 3D and 2D data, as shown by the common use of color jittering for data augmentation in 2D contrastive learning. Increasing texture diversity is a standard approach to improve the robustness of contrastive learning models.

## A.4 New Asset

Our training and evaluation code, along with the depth-specific text prompts, are included in the supplementary material. The generated depth-aligned images and the processed benchmark dataset will be made publicly available or provided to reviewers upon request for this submission.

## A.5 Additional Quantitative Analysis

**More Results for Zero-Shot Classification.** Table 7 presents the zero-shot classification results of OpenDlign on the long-tail Objaverse-LVIS [23] dataset. Remarkably, even the lightest OpenDlign variant (OpenDlign-B32) outperforms the previous state-of-the-art method by 8.3% in Top1 accuracy.

Table 7: Zero-shot classification results on Objaverse-LVIS [23]. The best-performing results are presented in bold, while the second-best results are underlined. Our models are highlighted in blue.

| Training | 3D Open-World | CLIP | Objaverse-LVIS [23] | | |
|---|---|---|---|---|---|
| Source | Methods | Variant | Top1 | Top3 | Top5 |
| 2D inferences | PointCLIP [14] | ResNet-50 | 1.9 | 4.1 | 5.8 |
| No Training | PointCLIP V2 [13] | ViT-B-16 | 4.7 | 9.5 | 12.9 |
| ShapeNet | CLIP2Point [15] | ViT-B-32 | 2.7 | 5.8 | 7.9 |
| | ULIP-PointBERT [17] | SLIP [50] | 6.2 | 13.6 | 17.9 |
| | OpenShape-SparseConv [18] | ViT-bigG-14 | 11.6 | 21.8 | 27.1 |
| | OpenShape-PointBERT [18] | ViT-bigG-14 | 10.8 | 20.2 | 25.0 |
| | TAMM-SparseConv [20] | ViT-bigG-14 | 13.6 | 24.2 | 29.3 |
| | TAMM-PointBERT [20] | ViT-bigG-14 | 13.7 | 24.2 | 29.2 |
| | OpenDlign-B32 | ViT-B-32 | 22.0 | 36.7 | 43.2 |
| | OpenDlign-B16 | ViT-B-16 | 25.7 | 42.2 | 48.8 |
| | OpenDlign-L | ViT-L-14 | 32.6 | 50.6 | 57.3 |
| | **OpenDlign** | **ViT-H-14** | **35.0** | **53.1** | **60.0** |

**More Results for Few-Shot Classification.** Table 8 presents the few-shot classification results of OpenDlign on the long-tail Objaverse-LVIS [23] dataset. OpenDlign consistently outperforms all baselines from 1-shot to 16-shot scenarios. Surprisingly, although OpenShape demonstrates higher zero-shot classification performance than ULIP, their few-shot results are poorer, indicating potential overfitting during multimodal alignment.

**More Results for Zero-Shot 3D Object Detection.** We followed the setting in [52] to compare our method with two 3D open-world approaches, focusing on object detection tasks OV-3DET [52] and CoDA [53] on the Scannet [51] dataset, as shown in Table 9. The results demonstrate that OpenDlign performs comparably to the state-of-the-art CoDA method, which is promising given that OpenDlign is not specifically designed for open-world 3D detection.

Table 8: Few-shot classification results on Objaverse-LVIS [23].

| Model | Objaverse-LVIS [23] | | | | |
| --- | --- | --- | --- | --- | --- |
| | 1-Shot | 2-Shot | 4-Shot | 8-Shot | 16-Shot |
| ULIP-PointBERT [17] | 13.3 | 16.3 | 20.3 | 25.3 | 32.3 |
| OpenShape-PointBERT [18] | 12.2 | 15.6 | 18.2 | 22.2 | 28.3 |
| OpenShape-SparseConv [18] | 13.4 | 16.9 | 19.7 | 22.9 | 28.9 |
| TAMM-PointBERT [20] | 13.5 | 18.0 | 22.4 | 27.3 | 33.6 |
| **OpenDlign (ours)** | **21.4** | **28.4** | **34.6** | **40.0** | **46.8** |

Table 9: Comparison of 3D open-vocabulary object detection methods in the same setting as OV-3DET on ScanNet. 'Mean' represents the average precision across all 20 categories.

| Methods | Mean | toilet | bed | chair | sofa | dresser | table | cabinet | bookshelf | pillow | sink |
| --- | --- | --- | --- | --- | --- | --- | --- | --- | --- | --- | --- |
| OV-3DET | 18.02 | 57.29 | 42.26 | 27.06 | 31.50 | 8.21 | 14.17 | 2.98 | 5.56 | 23.00 | 31.60 |
| CoDA | 19.32 | 68.09 | 44.04 | 28.72 | 44.57 | 3.41 | 20.23 | 5.32 | 0.03 | 27.95 | 45.26 |
| OpenDlign (Ours) | 19.27 | 58.30 | 41.95 | 34.13 | 32.25 | 12.49 | 18.12 | 3.89 | 4.15 | 25.41 | 34.53 |

| Methods | | bathtub | refrigerator | desk | nightstand | counter | door | curtain | box | lamp | bag |
| --- | --- | --- | --- | --- | --- | --- | --- | --- | --- | --- | --- |
| OV-3DET | | 56.28 | 10.99 | 19.27 | 0.77 | 0.31 | 9.59 | 10.53 | 3.78 | 2.11 | 2.71 |
| CoDA | | 50.51 | 6.55 | 12.42 | 15.15 | 0.68 | 7.95 | 0.01 | 2.94 | 0.51 | 2.02 |
| OpenDlign (Ours) | | 58.20 | 8.74 | 17.85 | 5.49 | 1.53 | 11.46 | 6.82 | 5.03 | 1.54 | 3.17 |

**Ablation Study on Fine-Tuning Strategies.** Table 10 demonstrates that fine-tuning only attention layers of 1 transformer block is adequate for OpenDlign to learn effective 3D representation. Moreover, placing the trainable block for fine-tuning in an independent branch (residual tuning) is crucial, as it preserves the strong capability of pre-trained CLIP in extracting robust image representations from depth-aligned images.

Table 10: Ablation study on various fine-tuning strategies. 'Residual Tuning' decides whether to tune the transformer blocks connected residually to the CLIP encoder. 'Trainable Copy' initializes the block randomly or with CLIP parameters. 'Block Count' is the number of trainable blocks.

| Residual Tuning | Trainable Copy | Block Count | $\mathcal{L}_{dist}$ | ModelNet40 [46] | | | ScanObjectNN [47] | | | OmniObject3D [48] | | |
| --- | --- | --- | --- | --- | --- | --- | --- | --- | --- | --- | --- | --- |
| | | | | Top1 | Top3 | Top5 | Top1 | Top3 | Top5 | Top1 | Top3 | Top5 |
| ✗ | ✓ | 1 | ✗ | 56.8 | 71.7 | 77.0 | 40.0 | 59.8 | 72.2 | 24.4 | 37.1 | 43.2 |
| ✓ | ✗ | 1 | ✗ | 81.8 | 96.0 | 98.0 | 57.9 | 76.4 | **84.4** | 30.9 | 45.6 | 51.8 |
| ✓ | ✓ | 1 | ✗ | 82.2 | 95.9 | 98.4 | 59.1 | 76.6 | 83.4 | 31.0 | 45.9 | 52.9 |
| ✓ | ✓ | 2 | ✓ | 82.3 | **96.5** | 98.4 | 58.7 | 76.6 | 83.9 | 30.8 | 46.2 | 52.8 |
| ✓ | ✓ | 3 | ✓ | 82.2 | 96.3 | 98.4 | 58.0 | 76.3 | 84.0 | 30.6 | 45.7 | 52.1 |
| ✓ | ✓ | 1 | ✓ | **82.6** | 96.2 | **98.4** | **59.5** | **76.8** | 83.7 | **31.3** | **46.7** | **53.2** |

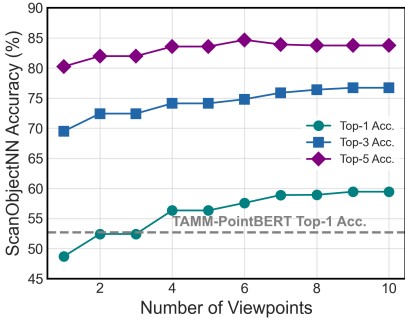

Figure 6: Effect of view count on OpenDlign's zero-shot performance on ScanObjectNN [47]

**More Results for Varying Number of Views.** Fig. 6 demonstrates that OpenDlign's zero-shot classification performance on the ScanObjectNN dataset generally improves as the number of views increases. Notably, OpenDlign surpasses the leading baseline model, TAMM-PointBERT [20], with approximately two views of depth maps. Table 11 presents detailed results of OpenDlign's performance on 16-shot classification with varying view counts across three benchmark datasets. These results further confirm that OpenDlign's few-shot performance, like its zero-shot performance, consistently improves with an increasing number of views.

Table 11: Effect of the number of viewpoints for 16-shot classification.

| Number of Viewpoints | 1 | 2 | 4 | 6 | 8 | 10 |
|---|---|---|---|---|---|---|
| ModelNet40 [46] | 74.3 | 78.3 | 84.6 | 86.1 | 87.4 | 87.6 |
| ScanObjectNN [47] | 46.0 | 50.5 | 60.6 | 68.8 | 75.6 | 79.0 |
| OmniObject3D [48] | 40.8 | 45.7 | 55.1 | 59.8 | 62.5 | 65.8 |

## A.6 Additional Qualitative Analysis

**Effect of Contour-Aware Projection.** Fig. 7 visually compares depth maps generated using PointCLIP V2's projection [13] technique and our contour-aware projection method. In the PointCLIP V2 projection, the projected maps are very blurry, losing some contour and shape details, as evidenced by the bed frame gaps being inaccurately filled. Conversely, our contour-aware projection preserves more of the original objects' contours and structures, accurately showing the bed frame gaps.

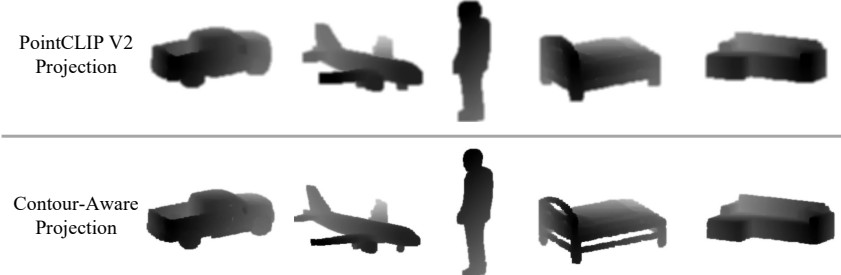

Figure 7: Comparison between depth maps from PointCLIP V2 [13] and contour-aware projection.

**More Visualizations of Cross-Modal Retrieval.** More examples of 3D shape retrieval that showcase the cross-modal retrieval capabilities of OpenDlign are illustrated in Fig. 8. The results demonstrate that the 3D shapes retrieved are in semantic alignment with the image and text queries.

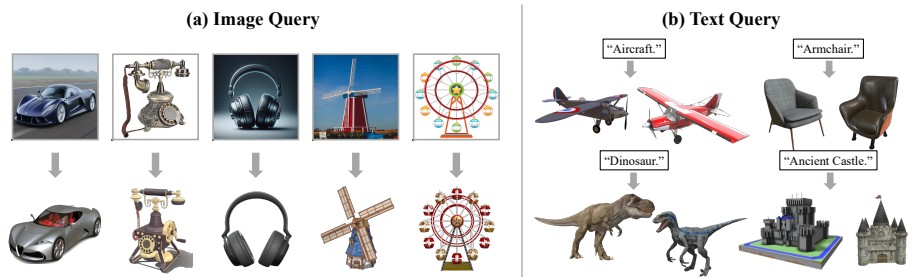

Figure 8: Additional 3D shape retrieval results. (a) Most similar 3D shapes for each image query. (b) Two most similar 3D shapes for each text query.

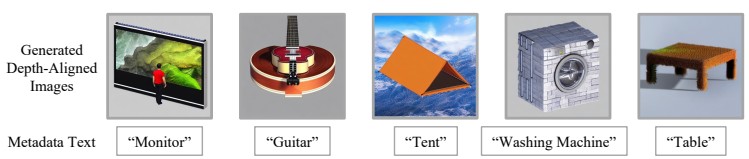

Figure 9: Examples of low-quality generated depth-aligned images.

**Examples of Low-Quality Depth-Aligned Images.** Fig. 9 showcases bad examples of depth-aligned images. For instance, the diffusion model may interpret 'monitor' as a person observing something, or it might create unrealistic images such as a washing machine made of stone. Additionally, it might depict a tent floating in the sky, which deviates from real-world expectations. These examples underscore the need to eliminate inferior depth-aligned images for more generalized alignment.

