# OpenReview forum: "OpenDlign: Open-World Point Cloud Understanding with Depth-Aligned Images"
_NeurIPS.cc/2024/Conference — NeurIPS 2024 poster_

### Official Review · Reviewer_kzvy · 2024-07-12

**Soundness:** 2
**Presentation:** 2
**Contribution:** 2
**Rating:** 5
**Confidence:** 4

**Summary:**

This paper focuses on 3D open-world learning on classification. To address the limitation caused by existing CAD-rendered images in open-world 3D learning, it proposes to generate depth-aligned images from point cloud projected depth maps and diffusion models.

**Strengths:**

1. Generating depth-aligned images to assist open-world 3d learning is interesting.
2.  The method is evaluated on multiple 3D classification tasks.

**Weaknesses:**

1. It seems that the generated textures among multiple views for a single object are not consistent. Would this largely influence the performance?
2. Generating images with diffusion models for image/3D understanding is a prevailing area [A]. The author should discuss existing works in the related works. Also, It turns out the method is not novel for me.
3. 3D understanding uses scene-level points in most downstream applications, please try on 3D scene-level datasets.

[A] Is synthetic data from generative models ready for image recognition?

**Questions:**

Please refer to the weakness part.

**Limitations:**

No.

---

> ### Author Rebuttal · Authors · 2024-08-06
>
> We thank the reviewer for the interesting questions.
>
> **Q1: Would inconsistent textures in multi-view images of a single object impact performance?**
>
> Answer: We believe that inconsistent textures in multi-view images improve model performance in 3D representation learning, rather than hinder it. Here are our supporting arguments and experiment evidence:
>
> **Supporting Arguments:**
>
> 1. **Effective Knowledge Transferring**: Inconsistent textures between multi-view generated images allow depth maps to align with diverse sets of RGB images. This diversity enhances the transfer of rich 2D knowledge embedded in CLIP to 3D representation learning.
>
> 2. **Invariant Feature Learning**: Generating geometry-consistent but texture-inconsistent multi-view images can be beneficial for learning 3D representations that emphasize object shape, size, and contour, which remain robust to texture variations. This approach makes sense because 3D data primarily provide geometric features rather than color or texture.
>
> 3. **Independent View Pairing**: Similar to methods like CLIP2Point, ULIP, and OpenShape, our OpenDlign pairs a single-view depth map with a single-view RGB image for contrastive learning, instead of using all multi-view images of a single object. This means each view is treated independently, eliminating the need to maintain texture consistency across views.
>
> 4. **Conventional Augmentation in Contrastive Learning**: Texture features are not considered robust not only in 3D data but also in 2D images, as evidenced by the widely used color jittering for data augmentation in 2D contrastive learning. Improving texture diversity is a conventional strategy to enhance the robustness of contrastive learning models.
>
> **Experiment Evidence:**
>
> Table 1 demonstrates that using generated images with inconsistent textures in (+dlign) settings systematically outperforms texture-consistent CAD-rendered images used by other methods without (+dlign). Furthermore, the ablation study in Table 5 shows that OpenDlign achieves a **3.8\%** and **6.8\%** accuracy improvement on the ModelNet and ScanObjectNN datasets, respectively, when using texture-inconsistent images compared to depth-rendCAD with consistent texture.
>
> **Q2: Using diffusion-generated images for 3D understanding is not novel.**
>
> Answer: We acknowledge the growing interest in using diffusion-generated images for 2D understanding. However, after carefully considering the feedback and reviewing the referenced work, we respectfully disagree with the assertion that our paper's novelty is similar to the referenced work. Below, we outline our reasons:
>
> 1. CLIP is designed for 2D representation learning, not 3D understanding. The referenced work [1] also focuses on enhancing CLIP's 2D image understanding using diffusion-generated images. In contrast, our approach transfers CLIP's 2D knowledge into 3D understanding, expanding its application to a new task. Currently, the use of diffusion-generated images in 3D understanding is largely unexplored, as shown by the papers citing [1].
>
> 2. The data scarcity challenge in 3D vision is much more severe than in 2D vision. State-of-the-art CLIP models for 2D understanding use billions of real-world image-text pairs, highlighting the substantial volume of pretraining data. The authors [1] recognize the need to explore if diffusion-generated images benefit larger CLIP models pre-trained on extensive datasets. However, the 3D datasets available for training 3D open-world models (i.e., '3D CLIP') are synthetic and contain less than a million samples. Our work mitigates this critical gap, emphasizing its broader impact and significance.
>
> 3. The contributions of this study are multifold, including contour-aware depth-map projection, a multimodal alignment framework, depth-aware prompt design, and a continual learning method to mitigate catastrophic forgetting. These innovations consistently result in significant performance improvements, as demonstrated in Table 4. The use of depth-aligned images is one of the key contributions, potentially benefiting all existing 3D open-world methods.
>
> 4. In [1], the generated images need only to be consistent with the text prompt. Our work, however, requires images to be consistent with both the object's pose and the text prompt, necessitating a more sophisticated generation process.
>
> Furthermore, we reviewed recent work utilizing diffusion models for various 3D understanding tasks. These studies [2,3] primarily use diffusion features to replace CLIP features for specific downstream applications, such as semantic segmentation [2,3], with very few explicitly generating images. In contrast, our work, similar to PointCLIP and OpenShape, focuses on learning robust 3D representations. These representations are versatile and can enhance various downstream tasks, including point cloud classification and 3D object detection, without being limited to specific applications. 3D classification is just one of the downstream tasks used to evaluate representation robustness.
>
> **Q3: 3D understanding uses scene-level points in most downstream applications, please try on 3D scene-level datasets.**
>
> Answer: Open-world 3D representation learning is a new field, mainly focusing on object-level understanding due to limited 3D scene data. Our setup fully follows existing methods, and we included additional experiments showing OpenDlign's performance on open-vocabulary 3D detection using the **scene-level ScanNet dataset**, as shown in Table 1 in the rebuttal pdf. As noted in Appendix A.1, we plan to extend our work to scene-level representation learning because CAD-rendered images for multimodal alignment are not always available for 3D scenes.
>
>
> [1] Is synthetic data from generative models ready for image recognition? (ICLR 2023)
>
> [2] Open-Vocabulary 3D Semantic Segmentation with Text-to-Image Diffusion Models. (Arxiv)
>
> [3] Open-vocabulary panoptic segmentation with text-to-image diffusion models.(CVPR 2023)

---

> > ### Comment · Reviewer_kzvy · 2024-08-12
> >
> > Thanks for the author's rebuttal. My concerns have been addressed, and I would like to raise my rating to a borderline accept. Please include all contents in the rebuttal to the next version.

---

### Official Review · Reviewer_XK9f · 2024-07-13

**Soundness:** 2
**Presentation:** 3
**Contribution:** 3
**Rating:** 5
**Confidence:** 3

**Summary:**

This paper proposes OpenDlign, a framework for depth-based 3D understanding by aligning depth-image features through training on generated depth-aligned images, which overcomes the limitations of training on CAD-rendered images. Using point cloud data from ShapeNet, a contour-aware projection method is introduced to produce dense depth maps from the point clouds. To prepare the training data, an off-the-shelf depth-controlled diffusion model is employed to generate the corresponding RGB images from the projected depth maps. Additionally, techniques such as depth-specific text prompts and logit aggregation are adopted to enhance performance. Experiments demonstrate state-of-the-art results in zero/few-shot 3D classification, zero-shot 3D object detection, and cross-modal retrieval tasks.

**Strengths:**

- The main idea of using generated depth-image pairs rather than CAD renderings to align a depth encoder with the image encoder is a reasonable design choice, enhancing generalization and robustness.
- The experiments are very comprehensive. The proposed method appears to achieve state-of-the-art results across all the evaluated tasks, which is quite impressive. Ablation studies are also very thorough, revealing the contribution of each design choice. However, since I am not an expert in these tasks, I cannot fully confirm the true status of this model's performance compared to other state-of-the-art methods.
- The paper is generally well-written and easy to follow.

**Weaknesses:**

- The motivations for the major design choices in this work are questionable. They appear to be driven purely by the evaluation benchmark (on where all methods use point cloud as the input, and some methods convert point cloud into depth later), rather than focusing on training a good encoder specifically for 3D understanding from depth maps. If this is the case, the title could better reflect this by referring to point cloud-image alignment rather than depth-image alignment. Specifically:
  - Why not use mesh to directly render depth maps? If the starting point is depth-image alignment, there is no need to convert point clouds into depth maps, which is only meaningful if the evaluation benchmarks are based on point cloud inputs. The availability of real depth maps renders one of the major contributions, the contour-aware projection method, completely irrelevant to the topic of this work.
  - Why not use real images and predict the depth maps using existing depth estimators, such as DepthAnything? This approach should significantly reduce the computation overhead of dataset creation and enable training on even more diverse data. Again, it seems that the current design choices are purely driven by the point cloud-based evaluation benchmark.
- In L152, the definition of inverse depth is incorrect. Inverse depth, also known as disparity, is 1/D. Depth ControlNet should take the normalized inverse depth (normalized into [0, 1]) as the input, rather than 1 - D.

**Questions:**

Please refer to the questions in the weaknesses section.

**Limitations:**

Limitations have been discussed. I don't see any potential negative societal impact in this work.

---

> ### Author Rebuttal · Authors · 2024-08-07
>
> We appreciate the reviewer's constructive questions and feedback.
>
> **Q1: The motivation of the study is not focused on training a good encoder specifically for 3D understanding from depth maps.**
>
> Answer:  Our primary motivation is to learn robust point cloud representations for open-vocabulary problems, similar to the goal of PointCLIP [1], as highlighted in its title, **"Point Cloud Understanding by CLIP"**. We train a depth encoder mainly to indirectly enhance representation learning from point clouds. While the reviewer's suggestions are beneficial for building a strong depth encoder specifically for depth map understanding, their benefits may be limited when the focus is on point cloud understanding.
>
> Additionally, we want to clarify any confusion caused by our paper title. By "depth-aligned images," we refer to images generated using depth map information, not the problem of "depth-image alignment." Our approach aims to align point cloud projected depth maps and images, thereby implicitly enhancing **point cloud-image alignment** robustness. Recent works like ULIP and OpenShape use the term "3D understanding" broadly without explicitly mentioning "point cloud," but their core objective, like ours, is to learn robust point cloud representations.
>
> **Q2: Why not use mesh to directly render depth maps instead of projecting depth maps from point clouds?**
>
> Answer: Generating depth maps from point clouds is preferred over using 3D meshes for several reasons:
>
> 1. **Real-Time Applicability:** The contour-aware projection method in OpenDlign generates depth maps directly from point clouds, making it more efficient than 3D rendering from meshes. Creating rendered depth maps involves a computationally expensive multi-step process: first, reconstructing the surface from the point cloud to create a 3D mesh, and then rendering the 3D mesh to obtain depth maps. This process introduces noise and makes rendered depth maps impractical for many real-time applications [1].
>
> 2. **Consistency with Related Works:** The projection of multi-view depth maps from point clouds is a well-established approach. Studies like PointCLIP, PointCLIP v2, and CLIP2Point also focus on proposing better point cloud-to-depth map projection methods. The motivation of these studies, similar to OpenDlign, is to enhance computational efficiency towards real-time point cloud understanding.
>
> 3. **Compatibility with Mainstream Methods:** Mainstream 3D understanding methods, such as PointNet and Point Transformer, operate directly on point clouds. OpenDlign's use of point clouds ensures compatibility and efficient processing with these methods.
>
> In summary, OpenDlign focuses on point cloud representation learning, not depth map representation learning. If our goal were 3D understanding from depth maps, we could use various methods to obtain them. However, since our focus is on point cloud-based 3D understanding, we constrain our depth maps to be projections from point clouds. We hope these explanations clarify our motivation and approach.
>
> **Q3: Why not use real images and predict the depth maps using existing depth estimators, such as DepthAnything?**
>
> Answer: Thank you for your insightful question. Using real images and pseudo-depth pairs can indeed increase data diversity. However, a key challenge is the domain discrepancy between depth maps predicted by estimators like DepthAnything and those projected from point clouds. This discrepancy arises because DepthAnything's depth maps are influenced by depth estimation errors, while point cloud-projected maps have noise from depth densification. Training with synthetic mono-depth from DepthAnything might limit the model's ability to generalize to point-cloud projected depth maps due to this domain shift.
>
> To address this, a potential approach is to pre-train the model with a large dataset of image-pseudo depth pairs and then fine-tune it on projected depth maps. Using multiple depth estimators, such as DepthAnything, MiDaS, and ZoeDepth, to generate pseudo depth can further enhance data diversity and improve robustness. It sounds like a promising direction for our future work.
>
> **Q4: The definition of inverse depth is incorrect.**
>
> Answer: Thank you for pointing that out. We indeed use normalized inverse depth as conditional control for image generation, as evidenced by the correctly generated images shown in Figure 2 in the rebuttal PDF. We will correct this in the revised manuscript.
>
> [1] PointCLIP: Point Cloud Understanding by CLIP (CVPR 2022)

---

> > ### Comment · Reviewer_XK9f · 2024-08-14
> >
> > Thank you for the rebuttal. I believe most of my concerns can be resolved if "point cloud understanding" is emphasized in the title and abstract. Otherwise, I am uncomfortable with the current title and abstract, as they are misleading; they clearly suggest that the focus is on depth understanding.
> >
> > As a side note regarding the "Real-Time Applicability" section: If the focus is on training a depth encoder, ShapeNet already provides ground truth meshes, and depth rendering can be extremely fast through simple rasterization. However, for point cloud understanding, the proposed method is reasonable, provided that "point cloud understanding" is clearly indicated in the title and abstract.

---

> > > ### Author Response · Authors · 2024-08-14
> > >
> > > Thank you for your feedback. We are glad to know that our rebuttal addresses your most concerns. We will certainly revise our title, abstract and introduction to better highlight "point cloud understanding" in the next version. We apologize again for any confusion caused by the text that may have suggested our focus was on depth understanding rather than point cloud understanding.

---

### Official Review · Reviewer_ky3c · 2024-07-13

**Soundness:** 3
**Presentation:** 3
**Contribution:** 3
**Rating:** 7
**Confidence:** 4

**Summary:**

The paper introduces **OpenDlign**, a novel framework for open-world 3D representation learning by leveraging depth-aligned images generated from a diffusion model. OpenDlign aims to enhance the realism and texture diversity in the 3D learning process, overcoming the limitations of CAD-rendered images. The method involves fine-tuning the CLIP image encoder with depth-aligned images, achieving superior performance in zero-shot and few-shot classification tasks. Experimental results demonstrate significant improvements over existing state-of-the-art models on various benchmarks like ModelNet40, ScanObjectNN, and OmniObject3D.

**Strengths:**

- **Innovative Approach**: The introduction of depth-aligned images generated from a diffusion model is a novel and effective way to enhance texture diversity and realism in 3D learning.
   &nbsp;
- **Significant Performance Gains**: The experimental results show substantial improvements in zero-shot and few-shot classification tasks, highlighting the effectiveness of the proposed method.
   &nbsp;
- **Detailed Methodology**: The paper provides a comprehensive description of the methodology, including the contour-aware projection method and the multimodal alignment framework.
   &nbsp;
- **Broader Impact Considerations**: The authors discuss both positive and negative societal impacts, demonstrating awareness of the potential implications of their work.
   &nbsp;
- **Reproducibility**: The paper includes sufficient details on the experimental setup and training process, enhancing the reproducibility of the results.

**Weaknesses:**

- **Limited Dataset for Depth-Aligned Images**: The generation of depth-aligned images is limited to the ShapeNet dataset, which might restrict the generalizability of the results to other datasets.
     &nbsp;
- **Computational Resources**: The approach requires significant computational resources for generating depth-aligned images and fine-tuning the model, which might not be easily accessible to all researchers.
     &nbsp;
- **Potential Biases**: The paper acknowledges biases in the CLIP and ControlNet models but does not provide detailed strategies for mitigating these biases in the generated depth-aligned images.

**Questions:**

- More applications: could the authors provide application study in other 3D OV Understand tasks like open-vocabulary 3D object detection methods[1, 2]
     &nbsp;
- Can the authors provide more details on the computational cost and feasibility of generating depth-aligned images for larger datasets?
     &nbsp;
- How do the authors plan to address the potential biases in the depth-aligned images generated by the diffusion model?
     &nbsp;
- Are there any plans to extend the approach to other types of 3D datasets beyond ShapeNet?
     &nbsp;
- Could the authors elaborate on the scalability of the method when using larger CLIP models or other vision-language models?

[1] Yuheng Lu, Chenfeng Xu, Xiaobao Wei, Xiaodong Xie, Masayoshi Tomizuka, Kurt Keutzer, and Shanghang Zhang. Open-vocabulary point-cloud object detection without 3d annotation. In CVPR, 2023. 1, 3.
[2] Yang Cao, Zeng Yihan, Hang Xu, and Dan Xu. Coda: Collaborative novel box discovery and cross-modal alignment for open-vocabulary 3d object detection. In NeurIPS, 2023

**Limitations:**

The authors have provided discussions about the limitations.

---

> ### Author Rebuttal · Authors · 2024-08-06
>
> We appreciate the reviewer's insightful feedback. We’re glad to hear that you found our work innovative, effective and reproducible.
>
> **W1: Limited Dataset for Depth-Aligned Images**
>
> Answer: Due to limited resources, our generated dataset is currently limited to ShapeNet, as discussed in the limitations section. However, post-submission, we generated more images from the ensemble dataset to enhance our model's generalization capabilities. The table for Reviewer ky3c shows that existing SOTA methods' performance keeps improving on all benchmark datasets as the dataset grows from **52K** to **90K** depth-aligned images.
>
> **W2 & Q2: Computational Resources and Feasibility for Generating Depth-Aligned Images on Large Datasets.**
>
> Answer: We understand the reviewer's concern about the computational constraints. Below we detail the computational resources necessary for generating depth-aligned images, discuss the feasibility of scaling our method to larger datasets, and confirm data accessibility:
>
> 1. **Memory Efficiency**: Multi-view depth-aligned images can be generated on any GPU with 24GB of memory using the ControlNet model, ensuring memory-efficiency.
>
> 2. **Time Efficiency**: Initially, generating images took two weeks using eight Quadro RTX 6000 GPUs (released in 2018). With newer RTX 4090 GPUs, which are 2.78 times faster, this process can now be completed in less than a week. The RTX 4090 is a consumer GPU, making it more accessible to researchers.
>
> | GPU Model       | Memory | Time/10-view Image |
> |-----------------|--------|--------------------|
> | Quadro RTX 6000 | 24 GB  | 39s                |
> | RTX 4090        | 24 GB  | 14s                |
>
> 3. **Comparison with Rendering**: Previous methods that generate CAD-rendered images using rendering tools (e.g., Blender) also demand considerable computational power and time. Therefore, both CAD-rendered images and depth-aligned image generation face similar computational constraints.
>
> 4. **Data Accessibility**: We will open-source the existing depth-aligned image dataset for ShapeNet, allowing direct access for the research community to avoid redundant generation.
>
> 5. **Faster ControlNet:** Recent work, like DMD2 [1], could serve as a faster alternative to ControlNet. DMD2 achieves high-quality conditional image generation more efficiently with much fewer inference steps compared to traditional methods like ControlNet. Preliminary results indicate that DMD2's four-step inference process for depth-conditioned image generation is 2.59 times faster than ControlNet, as shown below:
>
> | Model      | Time/10-view Image |
> |------------|--------------------|
> | ControlNet | 14s                |
> | DMD2       | 5.4s               |
>
> **W3 & Q3: Potential Biases in ControlNet.**
>
> Answer: Firstly, we require further experiments to quantify the effect of bias on the 3D representation robustness. One potential solution to mitigate bias is using a VLM (e.g., GPT-4V) or an image captioning model to evaluate generated image attributes like color, texture, and style. If the generated images have too many consistent attributes (like the same color), the diffusion model can be prompted to regenerate the images using different seeds. We will explore this direction in future work.
>
> **Q1: Application Study on Open-Vocabulary 3D object detection.**
>
> Answer: In section 4.3, we have demonstrated that OpenDlign outperforms PointCLIP and PointCLIP v2 by a large margin on the open-vocabulary 3D object detection task. Additionally, we followed the setting in [1] to compare our method with OV-3DET and CoDA on scannet dataset. Specifically, we use pseudo-box pre-trained 3DETR to locate objects and OpenDlign (ViT-H-14) to classify objects.
>
> The results below show that OpenDlign performs comparably to the SOTA CoDA method. This is already promising because our method is not specifically designed for OV 3D detection. For detailed results across all categories, please see Table 1 in the rebuttal PDF.
>
> | Methods   | mAP   | Chair | Sofa  | Dresser | Bathtub | Desk  |
> |-----------|-------|-------|-------|---------|---------|-------|
> | OV-3DET   | 18.02 | 27.06 | 31.50 | 8.21    | 56.28   | 19.27 |
> | CoDA      | 19.32 | 28.72 | 44.57 | 3.41    | 50.51   | 12.42 |
> | OpenDlign | 19.27 | 34.13 | 32.25 | 12.49   | 58.20   | 17.85 |
>
> **Q4: Plans for Extending to Other 3D Datasets?**
>
> Answer: Yes, we plan to first apply our method to the entire ensemble dataset for a more robust 3D shape understanding. Our next goal is to adapt our approach to 3D scene understanding, inspired by studies like 3D-Vista [2], which have successfully used CLIP for learning 3D scene representations by aligning point clouds, images, and text. We envision our method serving as a strong baseline for this task.
>
> **Q5: Scalability of OpenDlign on Larger CLIPs.**
>
> Answer: Table 1 in the main paper shows the zero-shot performance of OpenDlign with four CLIP backbones (B-32, B-16, L-14, H14). The short answer is better CLIP in 2D understanding leads to better OpenDlign in 3D understanding. We will elaborate on this relationship in the revised paper. The table below shows the model size and pre-training datasets, and 2D zero-shot classification averaged over 38 datasets of different CLIP backbones (https://github.com/mlfoundations/open_clip/blob/main/docs/openclip_results.csv). ViT-H-14 is one of the most powerful CLIP models so far. We can compare this result with the results in Table 1 to validate OpenDlign's scalability.
>
> | Model      | pretrained | params (M) | Avg acc (\%)|
> |------------|------------|------------| --------|
> | ViT-B-32   | datacomp_xl | 151.28 | 57.95 |
> | ViT-B-16   | datacomp_xl | 149.62 | 61.47 |
> | ViT-L-14   | datacomp_xl | 427.62 | 66.27 |
> | ViT-H-14-quickgelu   | dfn5b      | 986.11 | 69.61 |
>
>
> [1] Open-vocabulary point-cloud object detection without 3d annotation. (CVPR 2023)
>
> [2] 3d-vista: Pre-trained transformer for 3d vision and text alignment. (ICCV 2023)

---

> > ### Comment · Reviewer_ky3c · 2024-08-11
> > **Further reply**
> >
> > Thank you to the authors for the discussion and additional experiments provided during the rebuttal period; most of my concerns have been addressed,  I hope the authors can update the comparative experiments with OV-3DET and CoDA in the final version, as this knowledge will be beneficial to the related fields of the community. Totally speaking, I am inclined to accept this paper and decide to raise my rating to 7. Gook luck :)

---

### Official Review · Reviewer_tXWH · 2024-07-14

**Soundness:** 3
**Presentation:** 2
**Contribution:** 3
**Rating:** 5
**Confidence:** 3

**Summary:**

The paper introduces OpenDlign, a novel approach to 3D open-world learning. Traditional 3D learning models struggle with unseen categories and typically rely on CAD-rendered images, which lack texture diversity and realism, leading to suboptimal alignment and performance. OpenDlign leverages depth-aligned images generated from a diffusion model to enhance texture variation and realism. It seems OpenDlign achieves superior zero-shot and few-shot performance across diverse 3D tasks.

**Strengths:**

1. Overall the core idea of this work makes sense, i.e., instead of using CAD rendering, it uses an off-the-shelf diffusion model to convert 3D assets to 2D images, This increases the diversity of training and should be able to improve the generalisation to unseen data.

2. The performance of the proposed method seems strong.

**Weaknesses:**

1. Although the method shows a strong performance improvement when using ShapeNet for training, such a gain is not clear when using the "Ensemble" dataset for training. For example, in Table 1, TAMM-PointBERT (+dlign) is only higher than TAMM-PointBERT by 1.2% on ModelNet40 Top1, and even worse than on ModelNet40 Top5. Similarly,  when tested on the ScanObjectNN, OpenShape-SparseConv (+dlign) is worse than OpenShape-SparseConv over all metrics. Does it mean the proposed method is not important/beneficial when the training data is large and diverse enough?

2. Probably I missed something, but it seems the authors did not mention the name of the diffusion model they used in the paper. As shown in Figure 2, the "Multi-View Depth-Aligned Images" are also fed into CLIP Image Encoder. However, how to ensure multi-view consistency is still an open problem for diffusion models. Usually, such generated multi-view images will have obvious inconsistency. How did the authors address this problem? This might be a contributing factor to the performance drop mentioned in point 1, where methods show decreased performance when using OpenDlign.

3. Section 3.2 indicates that the proposed framework generates a single set of multi-view images for a given 3D asset. It seems logical to generate multiple sets of multi-view images for a 3D asset to enhance training diversity and reduce dependency on image appearance. Is there a specific rationale behind limiting the generation to a single set of multi-view images?

**Questions:**

Overall the reviewer is concerned about the point 1 and 2 mentioned in the Weaknesses. The reviewer may adjust the score based on the authors' response.

**Limitations:**

Yes

---

> ### Author Rebuttal · Authors · 2024-08-06
>
> We greatly appreciate the reviewer's insightful feedback and valuable recommendations.
>
> **Weakness 1**: Unclear performance gain on ensemble dataset.
>
> Answer: We understand the concern regarding the scalability of our methods on large datasets. Here are some explanations and additional experimental results to address this issue:
>
> 1. **Low Proportion of Depth-Aligned Images:**  The performance on the ensemble dataset is not significant because only 6\% of the images are depth-aligned (from ShapeNet), while the rest are CAD-rendered. It is impressive that even using a small portion of depth-aligned images improves the existing SOTA model's performance in most settings.
>
> 2. **Small Benchmark Datasets:** Training on the ensemble dataset typically results in larger performance gains on larger datasets like OmniObject3D compared to smaller ones like ScanObjectNN and ModelNet40. This is because the ensemble dataset offers a broader range of categories, which is more advantageous for larger datasets with more labels, as shown in Table 1 L.223.  Specifically, ModelNet40, ScanObjectNN, and OmniObject3D have **40**, **15**, and **216** categories respectively.
>
> 3. **Domain Shift in ScanObjectNN:** The performance drop on ScanObjectNN is due to its domain shift with other 3D datasets. ScanObjectNN, being the only real-world point-cloud dataset in our experiments, features more **sparse** and **noisy** point clouds compared to those sampled from 3D meshes (e.g., ModelNet40 and OmniObject3D). This visual difference is demonstrated in Figure 1 of the rebuttal PDF. This domain shift causes previous methods to struggle with improving performance on ScanObjectNN as well. For example, OpenShape-PointBERT increases Top-1 accuracy by 21\% on OmniObject3D but only by 0.9\% on ScanObjectNN, as shown in Table 1.
>
> 4. **Saturation of ModelNet40 Performance:** ModelNet40 is a relatively easy benchmark, and existing methods have already achieved high performance on this dataset. The 1.2\% improvement from TAMM-PointBERT to TAMM-PointBERT (+dlign) on ModelNet40 Top-1 accuracy is already more significant than the 0.6\% improvement of TAMM-PointBERT over OpenShape-PointBERT on the same metric.
>
> To further validate the benefits of depth-aligned images on a larger dataset, we conducted an additional ablation study. We increased the proportion of depth-aligned images in the training dataset from 6\% to 10\%, resulting in a total of 90K multi-view depth-aligned images for training TAMM-PointBERT and OpenShape-SparseConv. The results in the table below can be compared to Table 1 (L.223), where all scores improve, demonstrating that these methods achieve positive gains on all metrics when using more depth-aligned images for training.
>
> | 3D Open-World Model | ModelNet40 Top-1 | ModelNet40 Top-3 | ModelNet40 Top-5 | ScanObjectNN Top-1 | ScanObjectNN Top-3 | ScanObjectNN Top-5 | OmniObject3D Top-1 | OmniObject3D Top-3 | OmniObject3D Top-5 |
> |----------------------|------------------|------------------|------------------|---------------------|---------------------|---------------------|---------------------|---------------------|---------------------|
> | OpenShape-SparseConv (+dlign) | 85.8 (+2.4)     | 96.9 (+1.3)      | 98.4 (+0.6)      | 57.5 (+0.8)         | 79.0 (+0.1)         | 90.0 (+1.4)        | 35.0 (+1.3)         | 52.8 (+3.5)         | 59.6 (+2.2)         |
> | TAMM-PointBERT (+dlign)       | 86.7 (+1.7)     | 97.1 (+0.5)      | 98.5 (+0.4)      | 62.8 (+7.1)         | 83.1 (+2.4)         | 91.7 (+2.8)        | 38.8 (+1.7)         | 56.2 (+2.7)         | 62.8 (+1.0)         |
>
> However, we acknowledge that generating depth-aligned images for the entire ensemble dataset is time-consuming and computationally expensive, which will be our focus in future work.
>
> **Weakness 2**: What is the name of the diffusion model? How do you ensure multi-view consistency?
>
> Answer: The diffusion model used in our work is the ControlNet v1.1 [1] depth model (L.151), which allows us to use depth information to control the pose and shape of generated images.
>
> There is no multiview inconsistency in terms of geometry because we generate depth-aligned images view by view, each with its respective depth map. ControlNet's conditional control helps maintain the same shape and pose as the input depth map, and a text prompt ensures the generated images have consistent object identity. Since these depth maps come from the same 3D point cloud, they result in consistent depth-aligned images across different views. Figure 2 in the rebuttal PDF provides visual examples to further demonstrate this consistency. Multi-view consistency remains an open problem for diffusion models when generating multi-view images from a single-view input [2,3], rather than from multi-view inputs.
>
> While the generated depth-aligned images are geometry-consistent, they are not texture-consistent across views. Each view may have different textures, which we believe positively impacts generalizable representation learning, as discussed in the main paper.
>
> **Weakness 3**: Why not generate multiple sets of multi-view images for a 3D asset to enhance training diversity?
>
> Answer: Due to computational constraints, we only generated one image per depth map (view) for each 3D asset in our experiments. We agree that generating multiple sets of multi-view images using different random seeds for a 3D asset can further enhance training diversity. This will be considered in our future work.
>
>
>
> [1] ControlNet 1.1: https://github.com/lllyasviel/ControlNet-v1-1-nightly
>
> [2] Shi, Ruoxi, et al. "Zero123++: a single image to consistent multi-view diffusion base model."
>
> [3] Liu, Yuan, et al. "Syncdreamer: Generating multiview-consistent images from a single-view image."

---

> > ### Comment · Reviewer_tXWH · 2024-08-12
> >
> > Thanks for the authors' rebuttal. My concerns are mostly solved and hence I tend to accept this work.

---

> ### Author Response · Authors · 2024-08-12
>
> Thank you for your feedback. We're pleased to know that our rebuttal has addressed your most concerns. If this resolves the issues, we would greatly appreciate it if you could reconsider your rating. Otherwise, we are always happy to answer further questions that you may have.

---

### Author Rebuttal · Authors · 2024-08-07

We thank the reviewers for their detailed and thoughtful feedback on our submission. We are grateful that most reviewers appreciated the soundness of our model design and acknowledged its strong performance on various downstream 3D understanding tasks.

For each reviewer, we addressed their questions point by point in the threads below, but we want to highlight three main points that have caused confusion among reviewers:

1. The depth-aligned images in our approach are consistent with multi-view geometry but not with texture. This design choice aims to more effectively transfer rich 2D semantic knowledge from CLIP for learning robust 3D representations that are invariant to texture or color variations, focusing more on geometric features like object shape and contour.

2. Using depth-aligned images enhances model generalization, regardless of the size of the CLIP backbones or the 3D dataset. This is supported by additional experimental results in the rebuttal PDF and the results in Table 1 of the main paper.

3. The motivation of OpenDlign, similar to the pioneering study PointCLIP, focuses on learning robust 3D representations for point cloud understanding.

If any questions are not answered clearly, we are more than happy to provide further clarification in the discussion session.

---

### Decision · Program_Chairs · 2024-09-25

**Decision:**

Accept (poster)

**Comment:**

The paper leverages an off-the-shelf diffusion model to convert 3D assets into 2D images, increasing the diversity of training data and improving generalization to unseen data. Strong performance was demonstrated on the ModelNet40 and OmniObject3D datasets. Initially, the paper received one reject, two borderline acceptances, and one weak acceptance. The primary concerns raised were the scalability of the methods on large datasets, inconsistent textures in multi-view scenarios, and the novelty of using diffusion-generated images for 3D understanding. However, the authors addressed these concerns effectively in their rebuttal, providing additional experimental results and clarifying the paper's novelty. As a result, the final ratings were three borderline acceptances and one acceptance. Given the consistent shift towards acceptance, the Area Chair recommended the paper for acceptance.